# A Practical Algorithm for
# Feature-Rich, Non-Stationary Bandit Problems

**Wei Min Loh**                                              *wmloh@uwaterloo.ca*
*University of Waterloo, Vector Institute*

**Sajib Kumer Sinha**                                  *Sajib_Sinha@manulife.com*
*Manulife Financial*

**Ankur Agarwal**                                  *Ankur_Agarwal@manulife.com*
*Manulife Financial*

**Pascal Poupart**                                       *ppoupart@uwaterloo.ca*
*University of Waterloo, Vector Institute*

**Reviewed on OpenReview:** *https://openreview.net/forum?id=tRbwfej9uY*

## Abstract

Contextual bandits are incredibly useful in many practical problems. We go one step further by devising a more realistic problem that combines: (1) contextual bandits with dense arm features, (2) non-linear reward functions, and (3) a generalization of correlated bandits where reward distributions change over time but the degree of correlation maintains. This formulation lends itself to a wider set of applications such as recommendation tasks. To solve this problem, we introduce *conditionally coupled contextual* ($C_3$) Thompson sampling for Bernoulli bandits. It combines an improved Nadaraya-Watson estimator on an embedding space with Thompson sampling that allows online learning without retraining. Empirical results show that $C_3$ outperforms the next best algorithm by 5.7% lower average cumulative regret on four OpenML tabular datasets as well as demonstrating a 12.4% click lift on Microsoft News Dataset (MIND) compared to other algorithms.[1].

## 1 Introduction

Multi-armed bandits are applicable in many domains where there is high stochasticity and limited opportunities to fully explore all possible arms (Lattimore & Szepesvári, 2020). A more useful variant of the problem called *contextual bandit* (Lu et al., 2010) tackles a significantly harder problem where it aims to optimize for the best arm for a given context. Contextual bandits find applications in many domains including recommender systems, online advertising, dynamic pricing, and alternatives to A/B testing.

In several applications, the arms can be decomposed into a set of features such that different arms share some features and therefore their reward distributions may be dependent (which we refer to as *coupled* arms). Furthermore, the reward distributions of the arms may evolve over time, leading to non-stationarity. This paper focuses on non-stationary contextual bandits with coupled arms.

To motivate the investigation into coupled arms, or coupling in general, we consider an example of strong coupling in product recommendation. Complementary goods such as bicycles and helmets are typically strongly coupled. If the demand for bicycles rises, it is likely that the demand for helmets would go up too. Cycling, in some countries, is a seasonal activity where sales of bicycles and helmets differ during summer and winter.

---

[1]Our implementation can be found on GitHub (`https://github.com/wmloh/c3`).

This provides useful information for an agent when recommending products. Unlike time-series forecasting, we do not directly model the demand over a future time period. Instead, we capture features that might suggest coupling, then if one product has a high demand, it would immediately infer that a strongly coupled product would also have a high demand. This is beneficial for any bandit algorithm that has to balance exploration and exploitation of information.

**Main Contributions**   One of our contributions is introducing the notion of coupled arms that are ubiquitous in many practical applications. The primary contribution is developing an algorithm called *conditionally coupled contextual* ($C_3$) *Thompson sampling* that solves contextual bandits with correlated/coupled arms in bandit or recommendation tasks. To the best of our knowledge, it is the first algorithm that can solve contextual bandits with correlated/coupled arms in a non-stationary setting. Unlike many other neural contextual bandit approaches, there are no lengthy gradient-based updates at inference time. $C_3$ can also leverage arm features, which reduces the cold-start problem on arms, with the added benefit of working with a variable set of valid arms.

## 2   Related Works

### 2.1   Contextual Bandits

On the contextual bandit front, LinUCB by Li et al. (2010) can be considered one of the pioneering contextual bandit algorithms that demonstrated success through the use of simulated evaluation based on the Yahoo! news dataset. Chu et al. (2011) followed up with a rigorous theoretical analysis of a variant of LinUCB. The other popular paradigm is the Bayesian approach where Agrawal & Goyal (2013) developed a contextual bandit algorithm with Thompson sampling with a Gaussian likelihood and prior, assuming a linear payoff function. Unfortunately, there was no empirical evaluation on a practical problem.

With deep neural approaches on the rise, the contextual bandit community has been focusing on algorithms that can learn non-linear reward functions. One of the earlier algorithms was the KernelUCB by Valko et al. (2013) which extends LinUCB by exploiting reproducing kernel Hilbert space (RKHS). Similarly, Srinivas et al. (2009) generalized Gaussian processes for a contextual bandit setting by introducing the GP-UCB algorithm. While both algorithms attain a sublinear regret, practicality is limited since both have cubic time complexity in terms of the number of samples. After the breakthrough in the theoretical understanding of neural networks, particularly the neural tangent kernel (NTK) by Jacot et al. (2018), Zhou et al. (2020) developed NeuralUCB which is a neural network-based contextual bandit algorithm with a complete theoretical analysis and a suite of empirical analysis which outperforms many algorithms in tabular dataset benchmarks from OpenML.

More recent advancements include SquareCB (Foster & Rakhlin, 2020) which reduces the problem of contextual bandits to an online regression problem. Under mild conditions, SquareCB along with a black-box online regression oracle has optimal bounded regret with no overhead in runtime or memory requirements. While they work on most regression models, the performance is highly dependent on the quality of the selected oracle. Kveton et al. (2020) introduced their take on randomized algorithms for contextual bandits. Their novel contributions include an additive Gaussian noise for a bandit setting that can be introduced to complex models such as neural networks.

### 2.2   Other Relevant Bandits

Several specialized bandit algorithms may be relevant to our problem. Basu et al. (2021) introduced a variant called contextual blocking bandit that handles a variable set of arms but assumes the selected arms of an agent influence the future set of valid arms. Their work revolves around this idea but ultimately differs in having a fixed, finite set of overall arms.

The problem of non-stationary reward distributions in bandits is usually referred to as a restless bandit, which is proposed by Whittle (1988). Wang et al. (2020) introduced the Restless-UCB algorithm which provably solves restless bandits, but does not account for context in the environment. Chen & Hou (2024)

improves upon this by leveraging context and budget constraints. A specialized solution by Slivkins & Upfal (2008) assumes that reward distribution changes gradually and works by continuously exploring while sometimes following the best arm based on the last two observations. However, they assume that all arms are independent and can be modelled as a Brownian motion which is uncommon in practice.

In the realm of sleeping bandits where some arms are occasionally not valid, Slivkins (2011) introduced the contextual zooming algorithm that adaptively forms partitions in a similarity space. Operating on a space, as opposed to having fixed arms, allows them to effectively tackle the sleeping bandit problem. While they offer an innovative framework, it is likely that partitions in high-dimensional spaces (e.g. from large language models) would not be tractable.

Kleine Buening et al. (2024) proposed a situation that is potentially relevant to many applications from a game-theoretic perspective. Goals of agents could conflict – for example, an arm could be a video from a content creator on a public platform. In the current state, "clickbait" videos would maximize the reward for the content creator but are seen as a negative phenomenon overall on the platform. Kleine Buening et al. (2024) devised an incentive-aware learning algorithm to ensure that the learner obtains the right signal.

## 2.3 Recommender Systems

There is growing interest in applying contextual bandits on recommender systems (Ban et al., 2024). On the topic of recommender systems, the more prevalent form of recommendation models in recent years is typically based on neural networks Dong et al. (2022). A popular approach is the two-tower neural network employed by Huang et al. (2020) and Yi et al. (2019). A major benefit of such designs includes incredibly efficient retrieval within the learned embedding space as well as the ability to learn complex relationships of queries and items. Another design called BERT4Rec by Sun et al. (2019) leverages the power of transformers in sequential problems to provide recommendations. However, in the mentioned recommender designs, there is no element of exploration, unlike contextual bandits, resulting in a possibly greedy approach that may get stuck in a suboptimal policy. This can also be a major problem when user behaviour changes since all gradient updates to the models are based on historical data alone, and frequent retraining of the models can be expensive.

# 3 Problem Formulation

## 3.1 Contextual Bandit

In this paper, we focus on Bernoulli bandits where the conditional reward distribution is $R \sim$ Bernoulli($\mu(c,a)$) and $c \in \mathcal{C}$ is the current context. $\mathcal{C}$ is left to be an arbitrary space as long as it can be appropriately encoded. Each arm $a \in \mathcal{A}$ is a discrete class. The Bernoulli mean parameter $\mu : \mathcal{C} \times \mathcal{A} \to [0,1]$ is a continuous function whose value represents the probability of the reward being one. $\mu$ is assumed to be Lipschitz continuous in $\mathcal{C} \times \mathcal{A}$.

## 3.2 Coupled Arms: An Extension to Correlated Arms

Gupta et al. (2021) formulated the correlated multi-armed bandit problem where pulling an arm provides some information about another arm that is correlated. We view this as a special case of coupling.

Conditional reward distributions tend to be non-stationary in practice but we can still exploit some information on arms. While $\mu$ could change with time, there is an inherent structure to how $\mu$ of certain arms change. For example, the marginal probabilities of shoppers buying snowboards and skis are likely to be coupled, even though both probabilities would drop during the summer and rise during winter in a similar fashion.

In reference to Figure 1, up to the present, we see that the expected rewards for arms 1 and 2 are very similar and evolve similarly together in the time interval. We call these *strongly coupled arms.* Pulling one arm will give some information about the other arm in any time period, in contrast to correlated arms where

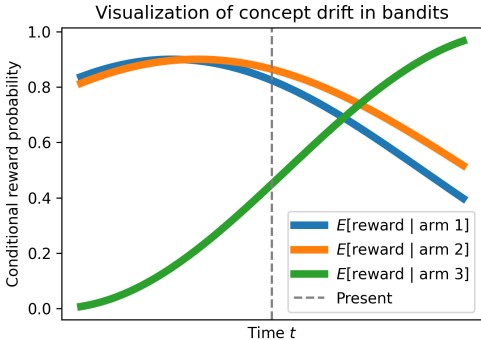

Figure 1: A three-arm example of strong and weak coupling during concept drift of expected reward distribution. Arms 1 and 2 are said to be *strongly coupled*, while arms 1 and 3 are said to be *weakly coupled*.

this condition is only true at a particular point in time. Conversely, arms 1 and 3 are *weakly coupled arms* since they have mostly different historical expected rewards.

Concept drift, in the context of bandits, can be defined as: there exist some times $t_1, t_2 \in \{1, 2, ..., T\}$ where $P(r|c, a, t_1) \neq P(r|c, a, t_2)$ ($r$ is the reward, $c$ is the context and $a$ is an arm) (Lu et al., 2018). The degree of coupling between arms $a$ and $a'$ for a given context $c$ is

$$\rho(a, a', c) = 1 - \frac{1}{T} \sum_{t=1}^{T} D_{\text{JS}}(P(r|c, a, t), P(r|c, a', t)) \tag{1}$$

where $D_{\text{JS}}$ is the Jensen-Shannon divergence over the reward distributions. Arms $a$ and $a'$ are said to be perfectly coupled for context $c$ if $\rho(a, a', c) = 1$.

### 3.3 Non-stationary Contextual Bandits with Coupled Arms

We extend vanilla contextual bandits to a more general problem. The Bernoulli mean parameter $\mu(c, a, t)$ is now a function of time too. We also assume that $\mu$ is Lipschitz continuous with respect to time.

Each arm $a \in \mathcal{A} \subseteq \mathbb{R}^d$ can be characterized with a vector of dense features, which implies that there are infinitely many possible arms but a finite number of arms are presented to an agent at each time step. We call them *valid arms* when they are presented to the agent at that particular time step. In a special case where arms do not have dense features, they can still be represented as one-hot encoded vectors.

In the presence of concept drift, i.e. $\mu(c, a, t) \neq \mu(c, a, t')$ generally for $t \neq t'$, and infinitely many possible arms, this can be a very difficult task. Here, we assume that there are strongly coupled arms that can be exploited. The degree of coupling is learnable from the arm features, conditioned on the context $c$.

### 3.4 Objective

The goal in both problems is to minimize the *cumulative regret* which is the cumulative difference between the reward of the best arm in hindsight $a^*$ and the reward of the chosen arm $a_t$ for a given context $c_t$ over all time steps (Lattimore & Szepesvári, 2020).

$$\text{CumulativeRegret}(T) = \sum_{t=1}^{T} \mu(c_t, a_t^*, t) - \mu(c_t, a_t, t) \tag{2}$$

## 4 Methodology

### 4.1 Embedding Model

This section pertains to the process of training an offline regression oracle, a class of optimization oracles for contextual bandits described by Bietti et al. (2021).

**Importance Weighted Kernel Regression**   The chosen approach to capture coupling between arms is based on the hypothesis that the empirical rewards of a relevant subset of reference samples can be used to estimate the reward of some unseen sample. Nadaraya-Watson kernel regression (NWKR) is a well-known nonparametric regression method (Nadaraya, 1964) that uses a weighted average of labels of neighbouring samples, weighted by a kernel function that satisfies some conditions.

Suppose there is a learnable space $\mathcal{S} \subset \mathbb{R}^d$ that represents some joint space of contexts $\mathcal{C}$ and arms $\mathcal{A}$, similar to the formulation by Slivkins (2011). We could use NWKR on some reference dataset $\mathcal{D}_{\text{ref}} = \{(s_i, r_i)\}_{i=1}^n$ containing historical context-arm embeddings $s_i \in \mathcal{S}$ and rewards $r_i \in \mathbb{R}$, with $\kappa : \mathcal{S} \times \mathcal{S} \to \mathbb{R}$ being the *radial basis function* (RBF) kernel to estimate the mean reward of an unseen sample $s \in \mathcal{S}$,

$$\hat{\mu}(s) = \frac{\sum_{s_i, r_i \in \mathcal{D}} \kappa(s, s_i) r_i}{\sum_{s_i \in \mathcal{D}} \kappa(s, s_i)} \tag{3}$$

Note that $\mu$ introduced in Section 3.3 is a function of context, arm, and time while $\hat{\mu}$ in Equation 3 is a function of context-arm (jointly) and reference dataset $\mathcal{D}$. Unlike time series algorithms, we do not directly model time as an independent variable since we have to make assumptions on how the reward distribution changes over time. Instead, we use historical examples in $\mathcal{D}$ that are close to the present time to indirectly condition on the time.

An issue with NWKR is the susceptibility to bias from drifts in the sampling distribution. The overall weight of samples may dominate the regression due to the disproportionately many samples in the vicinity of a query sample. This is particularly an issue in bandit algorithms where the distribution of arms selected will likely change as more data is ingested. The points in $\mathcal{S}$ provide information about $\mu$ in that subspace, but the frequency of points should not affect $\hat{\mu}$, except for a measure of confidence which will be discussed in Section 4.2. To mitigate sampling bias, we introduce *importance weights*. A sample is assigned a lower importance weight if it is located in the vicinity of many samples, and a higher importance weight otherwise. More precisely, the importance weight is defined as

$$w(s) = \frac{1}{\sum_{s_i \in \mathcal{D}} \kappa(s, s_i)} \in (0, 1] \tag{4}$$

The *importance weighted kernel regression* (IWKR) is defined as

$$\hat{\mu}(s) = \frac{\sum_{s_i, r_i \in \mathcal{D}} \kappa(s, s_i) w(s_i) r_i}{\sum_{s_i \in \mathcal{D}} \kappa(s, s_i) w(s_i)} \tag{5}$$

**Theorem 1.** *Suppose a vector of importance weights $\boldsymbol{w}$ of $n$ samples has been computed. The time complexity of updating the importance weights, given a new sample, is $\mathcal{O}(n)$.*

A naive implementation of the importance weights computation would incur a quadratic time complexity. However, this can be optimized to be linear time as shown in the proof in Supplementary Material A.1.

**Theorem 2.** *Suppose $\mu(s)$ is Lipschitz continuous on $\mathcal{S}$. In the limit of the size of the reference dataset $\mathcal{D}_{ref}$ where points are sufficiently sampled from the neighbourhood of some query point $s$, the importance weighted kernel regression with a radial basis function kernel is an approximate estimator of $\mu(s)$.*

The proof of Theorem 2 can be found in Supplementary Material A.2.

**Parametrization of Embedding Space**   While IWKR can estimate values, the input space may not be sufficiently calibrated with respect to the fixed kernel function. This can be rectified by training a multilayer perceptron as an embedding model $\phi : \mathcal{C} \times \mathcal{A} \to \mathcal{S}$ with IWKR towards a classification objective.

$$\min_{\phi} \mathbb{E}\left[\mathcal{L}_{\text{BCE}}(\hat{\mu}(\phi(c, a)), r) + \lambda \mathcal{L}_{\text{ECE}}(\hat{\mu}(\phi(c, a)), r)\right] \tag{6}$$

where $\mathcal{L}_{\text{BCE}}$ is the binary cross entropy loss and $\mathcal{L}_{\text{ECE}}$ is the expected calibration error (Naeini et al., 2015). Every context-arm pair will be embedded as $s = \phi(c, a)$ so that IWKR acts on an optimal space.

We incorporate calibration as an auxiliary objective to reduce overconfidence which is notoriously common in deep neural networks (Guo et al., 2017). In a bandit algorithm involving neural networks, calibration is important to avoid biases when facing a lack of data.

An optimal model would tightly cluster strongly coupled context-arm pairs. To encourage the learning of coupling in $\phi$, we can partition the reference dataset by time intervals $\mathcal{D}_{\text{ref}} = \mathcal{D}_{\text{ref}}^{(1)} \cup \cdots \cup \mathcal{D}_{\text{ref}}^{(T)}$ so that IWKR only uses samples from the relevant time interval only for a given query. This avoids averaging reward values from a different time period which may be subject to concept drift. The training process is described in Algorithm 1.

---

**Algorithm 1** $C_3$ training process

---

1: **Inputs**: Training dataset $\mathcal{D} = \{(c_i, a_i, r_i)\}_{i=1}^n$, neural network $\phi$
2: **for** epoch $e$ **do**
3:      Randomly split $\mathcal{D}$ into $\mathcal{D}_{\text{ref}} = (\boldsymbol{c}_{\text{ref}}, \boldsymbol{a}_{\text{ref}}, \boldsymbol{r}_{\text{ref}})$, and $\mathcal{D}_q$
4:      Embed reference $\boldsymbol{s} \leftarrow \phi(\boldsymbol{c}_{\text{ref}}, \boldsymbol{a}_{\text{ref}})$
5:      Compute importance weights $\boldsymbol{w}$ for $\boldsymbol{s}$
6:      **for** $(c, a, r) \in \mathcal{D}_q$ **do**
7:          Embed query $q \leftarrow \phi(c, a)$
8:          $\boldsymbol{s}', \boldsymbol{r}'_{\text{ref}} \leftarrow$ filter for samples in $\boldsymbol{s}, \boldsymbol{r}_{\text{ref}}$ such that they are in the same time interval as $q$
9:          Compute RBF weights between $\boldsymbol{s}'$ and $q$
10:         Estimate weighted mean reward $\hat{\mu}(q)$ using the RBF weights and $\boldsymbol{r}'_{\text{ref}}$
11:         Compute the sum of losses with $\hat{\mu}$ and $r$
12:         Perform gradient descent on $\phi$

---

**Algorithm Details**    The training is done batch-wise. The randomization in Step 3 forms a self-supervised learning objective by masking certain samples and creating a predictive subtask. $D_{\text{ref}}$ can be seen as the set of in-context samples and $D_q$ contains the training samples. All samples (both in $D_{\text{ref}}$ and $D_q$) are embedded with $\phi$, and the objective is to optimize the embedding space produced by $\phi$.

Step 9 computes the RBF weights between $q$ and every embedded reference sample in $\boldsymbol{s}'$. Then in Step 10, we apply Equation 5 on $q$ using the RBF weights, importance weights, and $\boldsymbol{r}'_{\text{ref}}$ to compute $\hat{\mu}(q)$. The gradient update should update $\phi$ to embed context-arm pairs with similar rewards closely.

## 4.2   Inference

This section extends the offline regression oracle by incorporating exploration with a Beta distribution and Thompson sampling.

**Thompson Sampling**    The embedding model with IWKR is trained towards a classification objective for predicting the expected reward. To incorporate an element of exploration, we adopt Thompson sampling. The conjugate prior of a Bernoulli bandit is a Beta distribution with parameters $\alpha$ and $\beta$, where $\alpha$ usually refers to the counts of $r = 1$. The notion of counts in a continuous embedding space $\mathcal{S}$ can be solved using partial counts of rewards weighted by the RBF kernel. However, this is complicated by importance weights since $\mathcal{S}$ was learned with IWKR.

The expected value of the Beta distribution should be impacted by $w(s)$ since it makes $\hat{\mu}$ less biased and robust against sampling distribution shifts. However, the variance of the Beta distribution should not be impacted by $w(s)$ since it would cause $\alpha$ and $\beta$ to lose information on the number of times the neighbourhood was sampled. A solution to this is to introduce some modifications to the computation of the parameters to

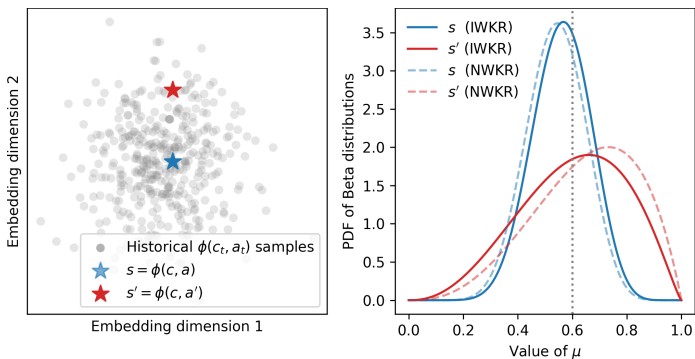

Figure 2: An example of Thompson sampling exploration in continuous spaces: [left] embedding space containing reference samples $\mathcal{D}_{\text{ref}}$ (circles) and different arms (stars) for a given context $c$, and [right] constructed Beta distribution with (IWKR) and without (NWKR) importance weights. The true $\mu$ of both arms for that context is 0.6.

the conjugate prior. Let $\eta(s) = \sum_{i=1}^{n} \kappa(s, s_i)$. Define

$$\alpha(s) := \eta(s) \frac{\sum_{i=1}^{n} \kappa(s, s_i) w(s_i) r_i}{\sum_{i=1}^{n} \kappa(s, s_i) w(s_i)} \qquad = \eta(s)\hat{\mu}(s) \tag{7}$$

$$\beta(s) := \eta(s) \frac{\sum_{i=1}^{n} \kappa(s, s_i) w(s_i)(1 - r_i)}{\sum_{i=1}^{n} \kappa(s, s_i) w(s_i)} \qquad = \eta(s)(1 - \hat{\mu}(s)) \tag{8}$$

With this, we can sample the posterior $\tilde{\mu}(s) \sim \text{Beta}(\alpha(s), \beta(s))$ from the resulting Beta distribution. The mean simplifies to

$$\mathbb{E}[\tilde{\mu}(s)] = \frac{\alpha(s)}{\alpha(s) + \beta(s)} = \hat{\mu}(s) \tag{9}$$

which is exactly IWKR. On the other hand, the information of the frequency the neighbourhood was sampled is still preserved because it can be shown that the total count is still a function of $n$,

$$\alpha(s) + \beta(s) = \sum_{i=1}^{n} \kappa(s, s_i) \tag{10}$$

The left side of Figure 2 illustrates two arms $a, a'$ when jointly embedded with context $c$. The gray circles refer to previously pulled arms $a_t$'s for different contexts $c_t$'s. For a new query with context $c$, arm $a$ (blue) is embedded closer to more samples hence it has essentially been pulled more often for context $c$. Using partial counts weighted by an RBF kernel, this manifests as a more peaked Beta distribution as shown on the right side, resulting in less exploration compared to $a'$ (red).

The combination of IWKR and Thompson sampling gives rise to *conditionally coupled contextual* Thompson sampling $(C_3)$. The term "conditionally" in $C_3$ refers to the degree of coupling being conditional on the context.

Figure 2 also shows the effectiveness of importance weights in a non-uniform sampling setting. The mean of distributions formed with IWKR is closer to 0.6 compared to NWKR, i.e. without importance weights.

**Approximate Bayesian Update**   Recall that IWKR is dependent on a reference dataset $\mathcal{D}$. The sampling of the posterior weighs every sample with the RBF kernel relative to some query point. This can be seen as a type of conditioning on a local subspace. After an arm is pulled and the reward is observed, we should have the triplet $(c_{n+1}, a_{n+1}, r_{n+1})$. Since the algorithm operates on $\mathcal{S}$ and requires the updated importance

Table 1: Comparison between algorithms where linear refers to both LinUCB, LinTS, and SquareCB while neural refers to both NeuralUCB and NeuralTS. $n$ refers to the number of samples seen.

| Algorithm | Inference time | Update time | Non-linear rewards | Non-stationary tasks | Arm features |
|---|---|---|---|---|---|
| $C_3$ | $\mathcal{O}(n)$ | $\mathcal{O}(1)$ | ✓ | ✓ | ✓ |
| Linear | $\mathcal{O}(1)$ | $\mathcal{O}(1)$ | ✗ | ✗ | partially |
| Neural | $\mathcal{O}(1)$ | $\mathcal{O}(n)$ | ✓ | ✗ | ✗ |

weights, to conserve time and memory, we can directly store the triplet $(\phi(c_{n+1}, a_{n+1}), r_{n+1}, w_{n+1}^{(n+1)})$ into $\mathcal{D}$. This is an approximate Bayesian update and is important for the online learning element.

Bayesian update typically assumes that every random variable in a sequence is identically distributed. The information gathered is directly stored in the parameter space of some statistical distribution, which will be updated using some closed-form algebraic expression. For $C_3$, the information is stored in the reference dataset $\mathcal{D}$ embedded on $\mathcal{S}$. This allows flexibility to both append and remove samples from $\mathcal{D}$. In problems with concept drift, the conditional reward distribution shifts as a function of time but a typical Bayesian update does not effectively handle this since it might simply average the distributions across time.

To mitigate the issues presented by non-stationarity without frequent retraining, $C_3$ allows the removal of older samples while appending the latest samples. Time can be seen as a special case of context and since $\mu$ is assumed to be Lipschitz continuous, samples nearer in the time dimension would be more relevant.

The entire inference pipeline can be summarized in Algorithm 2. The update process does not include any gradient updates, unlike many neural contextual bandit algorithms. The properties of $C_3$ and other contextual bandit algorithms are summarized in Table 1.

---

**Algorithm 2** $C_3$ inference process

---

1: **Inputs**: Reference dataset $\mathcal{D}_{\text{ref}} = (K, \boldsymbol{r}_{\text{ref}}, \boldsymbol{w})$, context $c$, set of valid arms $\{a_1, ..., a_k\}$, trained embedding model $\phi$
2: **for** each valid arm index $i \in [k]$ **do**
3:      Embed queries $q_i \leftarrow \phi(c, a_i)$
4:      Compute $\alpha(q_i)$, $\beta(q_i)$ with respect to $\mathcal{D}_{\text{ref}}$
5:      Sample $\hat{r}_i \sim \text{Beta}(\alpha(q_i), \beta(q_i))$
6: Play best arm $j \leftarrow \text{argmax}_{i \in [k]} \hat{r}_i$
7: Observe reward $r$
8: Append $(q_j, r)$ to $\mathcal{D}_{\text{ref}}$
9: Update $\boldsymbol{w}$ in $\mathcal{D}_{\text{ref}}$

---

**Theorem 3.** *Let the embedding space of $\phi$ be $S \subset [0, 1]^d$. Assume $\mu$ is L-Lipschitz. In a stationary bandit scenario, $C_3$ incurs an expected regret of*

$$\mathbb{E}[R_T] \in \mathcal{O}\left(L^{\frac{d}{d+2}} T^{\frac{d+1}{d+2}} (\log T)^{\frac{1}{d+2}}\right)$$

Proof of Theorem 3 can be found in Appendix A.3.

## 5 Experiments

Section 5.1 is the only experiment using synthetic data to demonstrate the hypothesis between coupling and embedding distance. Sections 5.2 and 5.3 use real-world datasets.

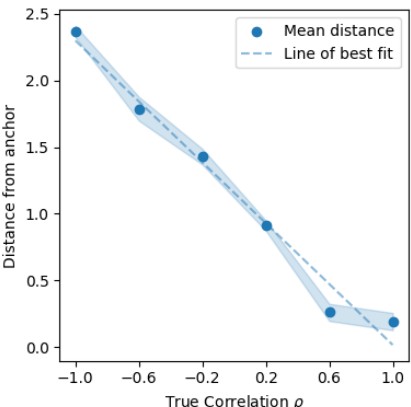

Figure 3: Distance from the anchor arm embedding as a function of correlation $\rho$ with 1.96 sigma error bars over 10 random seeds.

## 5.1 Coupled Arms Simulation

The following simulated example demonstrates that $\phi$ can capture the notion of coupling. Recall coupled arms generalize correlated arms by ensuring that correlation persists over time.

Suppose there is a set of arms $\{a_0, a_1, ..., a_6\}$. We call $a_0$ the *anchor arm* where the corresponding reward distribution is Bernoulli$(\mu_0)$. At time $t$, $\mu_0$ is randomly sampled from a Uniform$(0, 1)$ distribution. Then, the remaining $\mu_i$ for the rest of the arms are sampled such that they are either positively or negatively correlated with $\mu_0$. The chosen degree of correlation is fixed for any time for all arms with respect to the anchor arm. Arms can be sampled to obtain $(a_{it}, r_{it})$ pairs to learn $\mu_i$ until the end of the episode where this entire process repeats for time $t + 1$.

Complete details of the generation of coupled arms are described in Supplementary Material A.4. The chosen true correlations for arms $a_1, ..., a_6$ are -1.0, -0.6, -0.2, 0.2, 0.6, 1.0 respectively. For example, since $a_1$ is strongly negatively correlated to the anchor, if $\mu_0 = 0.9$ then it is very likely that $\mu_1 \approx 0.1$. On the other hand, $\mu_5$ would be in the vicinity of 0.9.

All $\mu_i$'s are sampled 200 times where for each time, 100 reward samples are collected. These reward samples are used to train $\phi$ using Algorithm 1 with $T = 200$ time intervals. A summary of arm embeddings is visualized in Figure 3. It follows the expectation where the more correlated $a_i$ is to $a_0$, the distance to $a_0$ is lower, and vice versa. To view one example of the spatial positioning of the arm embeddings in a scatter plot, see Supplementary Material A.6.

## 5.2 Contextual Bandit Experiments

This experiment demonstrates the efficacy of $C_3$ on the problem described in Section 3.1. In the footsteps of the work by Zhou et al. (2020), we evaluate using the same datasets from the OpenML repository by Vanschoren et al. (2014), namely `shuttle` (King et al., 1995), `MagicTelescope` (Bock, 2007), `covertype` (Blackard, 1998) and `mnist` (LeCun, 1998). For this set of experiments, the context space is the input space, and $\mathcal{A}$ is the corresponding label space. The reward is one if the selected arm matches the ground truth label, otherwise zero. Unlike Zhou et al. (2020), we do not standardize the inputs because we believe that gives the models some hindsight information which goes against the philosophy of multi-armed bandits. Note that this does not usually fit the typical setting of a bandit problem and $C_3$ targets a more general problem.

$C_3$ requires historical samples for $\phi$ to be trained, where historical samples are uncommon for bandit experiments but incredibly common for industry use cases. To ensure a fair comparison, we ensure that other baseline methods, such as LinUCB (Li et al., 2010), Thompson sampling for contextual bandits (LinTS) (Agrawal & Goyal, 2013), SquareCB (Foster & Rakhlin, 2020), NeuralUCB (Zhou et al., 2020), and neural

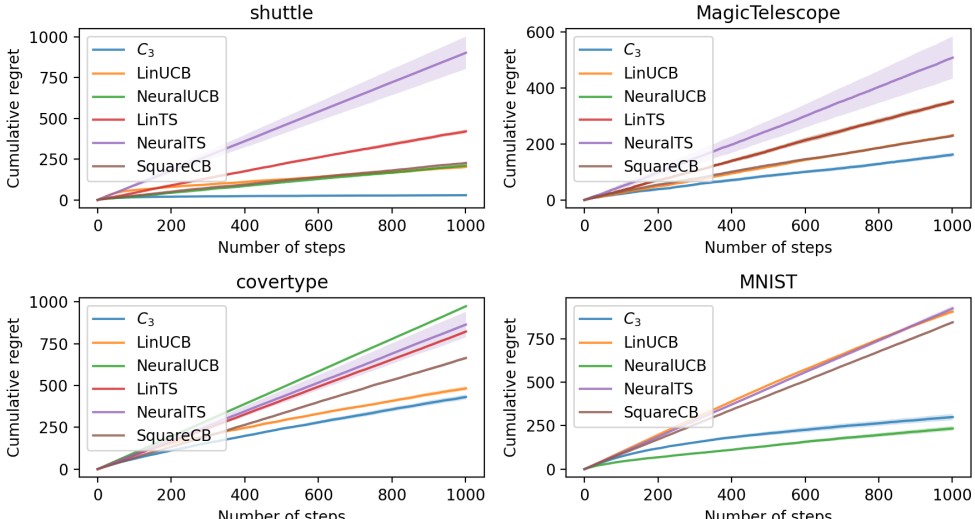

Figure 4: Cumulative regret of the test split of the four datasets with 1.96 sigma error bars over 10 random seeds. Note that in MNIST, LinTS cannot be computed due to numerical issues from the high dimensionality. In MagicTelescope, NeuralUCB and LinTS almost completely overlap because they all repeatedly exploit the same action after the initial steps.

Thompson sampling (NeuralTS) (Zhang et al., 2021), are given the same amount of information. All algorithms are updated using a subset for training and evaluated on a different test subset. This is necessary to avoid contamination when training $\phi$. The test split contains 1000 unseen samples. Appendix A.9 contains an ablation study of $C_3$ with different RBF bandwidth values.

The results are shown in Figure 4. $C_3$ outperforms most of the algorithms in most datasets. $C_3$ outperforms the next best algorithm for each dataset by 5.7% lower cumulative regret on average. In `shuttle`, `MagicTelescope`, and `covertype`, these problems are of lower dimensionality and more linearly separable hence LinUCB and SquareCB (with a linear regression oracle) perform well. On the other hand, NeuralUCB excels in the MNIST dataset since the multilayer perceptron works well with high-dimensional data.

## 5.3 News Recommendation

$C_3$ will be evaluated on news recommendation which is a realistic example for the problem described in Section 3.3. This paper uses the Microsoft News Dataset (MIND) by Wu et al. (2020) for evaluation. The context will be the frequency of a user's historical visits by news category. The arm space is the set of valid news articles to recommend. The reward is whether the user clicks the chosen news article. The objective of an algorithm is to minimize the cumulative regret. For clarity in results, regret is measured relative to the best performing algorithm where at time $t$, 0 is given to the best performing algorithm and the rest of the algorithms are given their respective relative regret.

We form dense representations of news articles from their titles using the pooler output of a BERT model (Devlin et al., 2019). We use principal component analysis to reduce the dimensions to 64 to be used as arm embeddings. Whenever a user visits, there is a small collection of possible articles to recommend, up to eight articles, but the valid arm set varies for each user.

In this set of experiments, $C_3$ will be configured so that every 100 steps, it will randomly remove approximately 20% of samples in $\mathcal{D}_{\mathrm{ref}}$ to account for the concept drift. The hypothesis is that because of the Lipschitz assumption, more recent samples would be more relevant in estimating the mean rewards.

Due to the rigidity of assumptions of other bandit algorithms tailored for theoretical results, some baseline algorithms in Section 5.2 could not effectively target all complexities of the problem for various reasons.

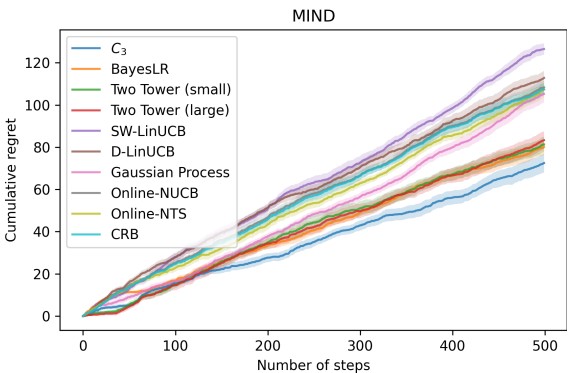

Figure 5: Cumulative regret of the MIND dataset with 1 sigma error bars over 10 random seeds. "small" and "large" refers to the relative number of parameters in the two tower models.

However, we add more specialized algorithms. We compare $C_3$ with one of the most popular recent designs for recommender systems: the two-tower neural network. Huang et al. (2020) uses a deep encoder for query information and another deep encoder for item information then uses the cosine similarity between the two embeddings. A Gaussian process with forgetting is also included Kaufmann et al. (2012), where the forgetting is necessary since it could not handle the entire history and needs to account for the non-stationarity. A similar baseline is using a Bayesian linear regression model that takes a concatenated vector of user and article features and returns the mean and standard deviation of estimated rewards, similar to the Gaussian process. Linear and neural baselines from Section 5.2 are modified to handle the non-stationarity elements. We also compare with the contextual restless bandit algorithm Chen & Hou (2024). Full experimental details are found in Supplementary Material A.11.

The result can be seen in Figure 5. $C_3$ demonstrates a click lift of 12.4% compared to the baselines. Initially, $C_3$ does not do as well as the two tower approaches (which are static). As the drift of click rate increases in magnitude, $C_3$ begins to adapt to the drift as it removes older samples in $\mathcal{D}_{\mathrm{ref}}$ while the other algorithms incur regret at the same rate.

## 6 Discussions

**Effective Data Utilization**  Unlike typical contextual bandit settings, $C_3$ requires the use of historical data to "warm-start" the algorithm. In reality, advertisement campaigns, pricing schemes, etc. will involve some degree of human-designed policy at the initial stages which means there could be some data albeit possibly sub-optimal. Sub-optimality is not an issue for the training of $\phi$ since the main objective of $\phi$ is to learn reward distributions and coupling, not optimality. As a result, $\phi$ can effectively utilize samples from prior campaigns or trials in an off-policy manner. Furthermore, $\phi$ can be resilient to concept drifts so data from a different time period may still be utilized.

**Generalization to Embedding Models**  This paper demonstrates that Thompson sampling acting on an embedding space can offer a method of exploration. However, only a simple multi-layer perceptron is used as an embedding model. There should be no restrictions on the model design or inputs as long as it is a cost-sensitive regression model. A side effect of operating on an embedding model is the ability to visualize the learned embedding space which can be useful for applications that require some transparency/explainability.

**Limitations of $C_3$**  The transductive learning aspect and importance weight updates can result in high numerical instability since it relies on many sum and division operations of floating points. This effectively disallows quantization to be used. Also, as an algorithm that relies on a dataset for inference, it may not scale to millions of points without some type of sampling if speed is crucial in the use case. The hyperparameter tuning of $\phi$ can be slightly challenging because the RBF kernel used in IWKR can result in vanishing gradients if points are too near or far from some query point, so the choice of bandwidth of the RBF kernel

is important. More crucially, the performance of $C_3$ is dependent on heuristics and whether the learned embedding space is well-calibrated.

## 7 Conclusion and Future Works

The design of the $C_3$ algorithm sets an applied perspective of using a contextual bandit algorithm on bandits and recommendation problems. Contextual bandit algorithms have built-in exploration and online learning components while recommender systems have deep encoders that scale well with high dimensional data. By combining the best of worlds, we gain several advantages in practice such as the ability to handle non-stationarity from concept drift, no retraining needed, and leveraging arm features. While this work contributes to the practical side of bandit algorithms, future works should include obtaining a non-stationary bound that extends the stationary regret bound in Theorem 3.

## Broader Impact Statement

To be best of our knowledge, our work does not have a direct negative impact as it outlines an algorithm to dynamically learn patterns. External factors such as the data by others on the $C_3$ algorithm, or the application of the algorithm on a malicious task would be out of our control.

### Acknowledgments

We thank Manulife Financial for the funding, guidance and exposure that motivated this work. Resources used in preparing this research were provided, in part, by the Province of Ontario, the Government of Canada through CIFAR, companies sponsoring the Vector Institute https://vectorinstitute.ai/partnerships/current-partners/, the Natural Sciences and Engineering Council of Canada and a grant from IITP & MSIT of Korea (No. RS-2024-00457882, AI Research Hub Project).

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

# A   Appendix

## A.1   Memoization of Importance Weight Computation

**Theorem 1** Suppose a vector of importance weights $\boldsymbol{w}$ of $n$ samples has been computed. The time complexity of updating the importance weights, given a new sample, is $\mathcal{O}(n)$.

*Proof.* Suppose there are $n$ samples in the embedding space $\{s_i \in \mathcal{S} : i \in [n]\}$. Consider the kernel matrix $L \in \mathbb{R}^{n \times n}$ which holds all pairwise RBF kernel values between every sample. From Equation 4, we can deduce that the sum of the $i^{\text{th}}$ row of matrix $L$ will be $w(s_i)$, and similarly for the sum of the $i^{\text{th}}$ column since $L$ is symmetric. The importance weight of the initial reference dataset $\mathcal{D}_{\text{ref}}$ can be calculated this way which takes on average $\mathcal{O}(n)$ per sample.

Suppose there is already the vector of importance weights for all samples in $\mathcal{D}_{\text{ref}}$ denoted as $\boldsymbol{w}^{(n)} = [w_1^{(n)} \cdots w_n^{(n)}] \in [0,1]^n$. We want to obtain an efficient update equation for $\boldsymbol{w}^{(n+1)}$. Naively computing $L$ with a new sample will result in a $\mathcal{O}(n^2)$ time update. To update efficiently, memoization would be useful since $w_i^{(n)}$ itself stores the reciprocal of a sum. During inference, the unweighted RBF similarity score will need to be computed. This result can be stored, and is denoted as $\boldsymbol{z}^{(n+1)} = [\kappa(s_{n+1}, s_1) \cdots \kappa(s_{n+1}, s_n)]$.

There are two steps here: update the existing $w_i^{(n)}$ for $i \in [n]$ and append $w_{n+1}^{(n+1)}$ to the new vector. To obtain $w_i^{(n+1)}$ for $i \in [n]$, the update equation can be expressed as a function of its previous value

$$
\begin{aligned}
w_i^{(n+1)} &= \frac{1}{\sum_{j=1}^{n+1} \kappa(s_i, s_j)} \\
&= \frac{1}{\kappa(s_i, s_{n+1}) + \sum_{j=1}^{n} \kappa(s_i, s_j)} \\
&= \frac{1}{z_i^{(n+1)} + \frac{1}{w_i^{(n)}}}
\end{aligned}
$$

which is a constant time operation for each $i \in [n]$ since all of the required values have already been computed. To compute the new importance weight,

$$
\begin{aligned}
w_{n+1}^{(n+1)} &= \frac{1}{\sum_{j=1}^{n+1} \kappa(s_{n+1}, s_j)} \\
&= \frac{1}{\kappa(s_{n+1}, s_{n+1}) + \sum_{j=1}^{n} \kappa(s_{n+1}, s_j)} \\
&= \frac{1}{1 + \sum_{j=1}^{n} z_j^{(n+1)}}
\end{aligned}
$$

which is an $\mathcal{O}(n)$ time operation for the new sample. Therefore, the entire update equation for the importance weight given a new sample is a linear time operation. $\qquad\square$

## A.2  Proof of Importance Weighted Kernel Regression Approximation

**Theorem 2** Suppose $\mu(s)$ is Lipschitz continuous on $\mathcal{S}$. In the limit of the size of the reference dataset $\mathcal{D}_{\mathrm{ref}}$ where points are sufficiently sampled from the neighbourhood of some query point $s$, the importance weighted kernel regression with a radial basis function kernel is an approximate estimator of $\mu(s)$.

*Proof.* The importance weighted kernel regression estimator of $\mu$ with a RBF kernel is defined in Equation 5. As the number of samples approaches $n \to \infty$ over a space centered at $s$, the importance weight converges to the inverse of the sampling distribution. The effective contribution of each neighbouring subspace becomes approximately uniform and the estimate becomes

$$
\begin{aligned}
\hat{\mu}(s) &= \frac{\int_{\mathcal{S}} \kappa(s, s') R(s') \, \mathrm{d}s'}{\int_{\mathcal{S}} \kappa(s, s') \, \mathrm{d}s'} \\
&= \int_{\mathcal{S}} \frac{\kappa(s, s')}{\int_{\mathcal{S}} \kappa(s, s'') \, \mathrm{d}s''} R(s') \, \mathrm{d}s'
\end{aligned}
$$

Now, we focus on the fractional term and show that it is simply the density of a Gaussian distribution.

$$
\begin{aligned}
\frac{\kappa(s, s')}{\int_{\mathcal{S}} \kappa(s, s'') \, \mathrm{d}s''} &= \frac{\exp\left(-\frac{\|s - s'\|}{2\sigma^2}\right)}{\int_{\mathcal{S}} \exp\left(-\frac{\|s - s''\|}{2\sigma^2}\right) \, \mathrm{d}s''} \\
&= \exp\left(-\frac{\|s - s'\|}{2\sigma^2}\right) \cdot \left((2\pi)^{-\frac{d}{2}} |\sigma|\right) \\
&= \Pr(X = s') \quad \text{where } X \sim \mathcal{N}(s, \sigma^2 I)
\end{aligned}
$$

Since we know that the conditional reward distribution is defined as $R \sim \mathrm{Bernoulli}(\mu(s))$, for a sufficiently small RBF kernel bandwidth $\sigma$, under the Lipschitz continuity assumption,

$$
\begin{aligned}
\hat{\mu}(s) &= \int_S P(X = s') R(s') \, \mathrm{d}s' \\
&\approx \mathbb{E}[R' | S = s] \\
&= \mu(s)
\end{aligned}
$$

$\qquad\square$

## A.3  Proof of Regret Bound in Theorem 3

For a fixed embedding map $\phi(c, a)$ trained offline, $S \subset [0, 1]^d$ is the learned joint embedding space, and we analyze directly on $S$. Assume that the true mean reward function $\mu : S \to [0, 1]$ is Lipschitz. There exists

$L > 0$ such that for all $s, s' \in S$,

$$|\mu(s) - \mu(s')| \leq L\|s - s'\|$$

Cumulative regret is formalized as follows. Let $s_t^* = \operatorname{argmax}_{s \in S_{A_t}} \mu(s)$ be an optimal action, where $S_A$ is the set of embeddings of all valid arms at time $t$. The expected cumulative regret is

$$R_T = \sum_{t=1}^{T} \mu(s_t^*) - \mu(s_t)$$

We standardize the notions. The RBF kernel $\kappa_h(\cdot, \cdot)$ has bandwidth $h > 0$. The reference dataset $D_{t-1} = \{(s_i, r_i)\}_{i=1}^{t-1}$ stores all embedding-reward tuples up to time $t-1$. The values of the Beta parameters $\alpha_{t-1}(s)$ and $\beta_{t-1}(s)$ are obtained by comparing a query embedding $s \in S$ to every embedding in the reference dataset, and their sum is the kernel mass

$$\eta_{t-1}(s) = \sum_{i=1}^{t-1} \kappa_h(s, s_i)$$

The IWKR procedure samples from the constructed Beta distribution

$$\Theta_t(s) \sim \mathrm{Beta}(\alpha_{t-1}(s), \beta_{t-1}(s))$$

and plays $s_t = \operatorname{argmax}_{s \in S_{A_t}} \Theta_t(s)$.

For the analysis of $C_3$, we use a truncated RBF kernel (compact support) for exact locality.

$$\kappa_h(s, s') := \exp\left(-\frac{\|s - s'\|^2}{2h^2}\right) \mathbb{1}\{\|s - s'\| \leq h\}$$

*Proof.* Let $S \subseteq [0, 1]^d$ without loss of generality by rescaling if necessary. Partition $[0, 1]^d$ into axis-aligned hypercubes (cells) of length $h$. Let $\mathcal{G}$ denote the set of all such cells intersecting $S$. The number of cells satisfies

$$|\mathcal{G}| \leq \left\lceil \frac{1}{h} \right\rceil^d \leq \left(\frac{2}{h}\right)^d \quad \text{for } h \leq 1 \tag{11}$$

For each cell $g \in \mathcal{G}$, fix a representative point $x_g \in g \cap S$, i.e. any point in the intersection. Define the cell index of any point $s \in S$ as $g(s) \in \mathcal{G}$, and the best cell representative $g^*$ as

$$g^* = \operatorname*{argmax}_{g \in \mathcal{G}} \mu(x_g)$$

Because $s^* \in S$ lies in some cell $g(s^*)$, the cell representative $x_{g(s^*)}$ is at distance of at most the cell diameter

$$\|s^* - x_{g(s^*)}\| \leq \sqrt{d}h$$

By the Lipschitz property,

$$\mu(s^*) - \mu(x_{g(s^*)}) \leq L\sqrt{d}h$$

Since $g^*$ maximizes $\mu(x_g)$ over $g$, we have $\mu(x_{g^*}) \geq \mu(x_{g(s^*)})$. Hence,

$$\mu(s^*) - \mu(x_{g^*}) \leq \mu(s^*) - \mu(x_{g(s^*)}) \leq L\sqrt{d}h$$

Decomposing regret gives

$$\mu(s^*) - \mu(s_t) = (\mu(s^*) - \mu(x_{g^*})) + (\mu(x_{g^*}) - \mu(s_t))$$
$$\leq L\sqrt{d}h + (\mu(x_{g^*}) - \mu(s_t)) \tag{12}$$

Up to this point, we approximated the continuum optimum $s^*$ to the best cell representation at a regret cost of $O(Lh)$. The next step is to bound the second term, which is the regret relative to the best cell representation. To do so, we exploit the locality of the truncated kernel.

For any query $s$, the kernel sums only involve past points $s_i$ from the same cell $g(s)$. Let the number of times we have played in cell $g$ up to time $t - 1$ be

$$N_g(t-1) := |\{i \leq t - 1 : g(s_i) = g\}|$$

The cell's respective cumulative binary reward counts are

$$S_g(t-1) := \sum_{i \leq t-1 : g(s_i) = g} r_i \qquad\qquad F_g(t-1) := N_g(t-1) - S_g(t-1)$$

Now, fix a query $s \in g$. The kernel mass is

$$\eta_{t-1}(s) = \sum_{i=1}^{t-1} \kappa_h(s, s_i) = \sum_{i:g(s_i)=g} \kappa_h(s, s_i)$$

Moreover, within a cell $g$, any two points have distance at most $\sqrt{d}h$, hence for $s, s_i \in g$,

$$\exp\left(-\frac{\|s - s_i\|^2}{2h^2}\right) \geq \exp\left(-\frac{d}{2}\right)$$

We call the right hand side $c_d \in (0, 1)$. Therefore,

$$c_d N_g(t-1) \leq \eta_{t-1}(s) \leq N_g(t-1) \tag{13}$$

**Lemma 1.** *Let $X \sim Beta(\alpha, \beta)$ with $\alpha, \beta \geq 1$, and let $p = \alpha/(\alpha + \beta)$. Then for any $\varepsilon \in (0, 1 - p)$,*

$$\Pr(X \geq p + \varepsilon) \leq \exp(-2(\alpha + \beta - 1)\varepsilon^2)$$

*Similarly, for any $\varepsilon \in (0, p)$,*

$$\Pr(X \leq p - \varepsilon) \leq \exp(-2(\alpha + \beta - 1)\varepsilon^2)$$

*Proof.* Let $X \sim \text{Beta}(\alpha, \beta)$ and $Y \sim \text{Binomial}(n, x)$ where $n = \alpha + \beta - 1$. By writing the regularized incomplete beta function representation of the Beta CDF and comparing it to the Binomial tail expression, one can obtain

$$\Pr(X \geq x) = \Pr(Y \leq \alpha - 1)$$

Take $x = p + \varepsilon$. Then $\mathbb{E}[Y] = n(p + \varepsilon)$. Note that $p \leq 1$, thus $\alpha - 1 \leq np$ because

$$np = (\alpha + \beta - 1)\frac{\alpha}{\alpha + \beta}$$
$$= \alpha - \frac{\alpha}{\alpha + \beta}$$
$$= \alpha - p$$
$$\geq \alpha - 1$$

Therefore, the event $Y \leq \alpha - 1$ implies $Y \leq np$

$$\Pr(X \geq p + \varepsilon) = \Pr(Y \leq \alpha - 1) \leq \Pr(Y \leq np)$$

Note that $np = n(p + \varepsilon) - n\varepsilon = \mathbb{E}[Y] - n\varepsilon$. Applying the Hoeffding/Chernoff lower-tail bound for Binomial random variables gives

$$\Pr(Y \leq \mathbb{E}[Y] - n\varepsilon) \leq \exp(-2n\varepsilon^2) = \exp(-2(\alpha + \beta - 1)\varepsilon^2)$$

Connecting both gives

$$\Pr(X \geq p + \varepsilon) \leq \Pr(Y \leq np) = \Pr(Y \leq \mathbb{E}[Y] - n\varepsilon) \leq \exp(-2(\alpha + \beta - 1)\varepsilon^2)$$

$\square$

Define the gap of a cell representative

$$\Delta_g := \mu(x_{g^*}) - \mu(x_g) \geq 0$$

Cells with $\Delta_g = 0$ are optimal within the grid. We want to show that each suboptimal cell $g$ is selected only $O(\log T / \Delta_g^2)$ times in expectation. Fix a suboptimal cell $g$ with $\Delta_g > 0$. Consider the event at time $t$ that IWKR selects a point in cell $g$. This can happen only if the Beta sample for cell $g$ is unusually high or the Beta sample for the best cell $g^*$ is unusually low.

Concretely, a good event refers to

$$E_g := \left\{ \left| \frac{S_g}{N_g} - \mu(x_g) \right| \leq \frac{\Delta_g}{4} \right\}$$

and similarly

$$E_{g^*} := \left\{ \left| \frac{S_{g^*}}{N_{g^*}} - \mu(x_{g^*}) \right| \leq \frac{\Delta_g}{4} \right\}$$

By Hoeffding's inequality for Bernoulli sums, conditional on $N_g = n$,

$$\Pr(E_g^c | N_g = n) \leq 2 \exp\left( -2n \left( \frac{\Delta_g}{4} \right)^2 \right) = 2 \exp\left( -\frac{n\Delta_g^2}{8} \right)$$

Let $\Theta_{t,g}$ be the cell-level random variable for the Beta sample corresponding to any query point in cell $g$ at time $t$. They are all functions of the same within-cell data, only differing in constants via kernel weights. Its parameters satisfy $\alpha + \beta = \eta$ and with Equation 13, we have $\eta \geq c_d N_g$.

On event $E_g$, the empirical mean is close to $\mu(x_g)$ and because the posterior mean is equal to the IWKR mean, which (within a cell) is a weighted average of the $r_i$'s, we can bound the posterior mean deviation by the same scale (absorbing kernel-weight constants into a dimension constant). Denoting $\mathcal{F}_{t-1}$ as the history up to $t - 1$,

$$|\mathbb{E}[\Theta_{t,g} | \mathcal{F}_{t-1}] - \mu(x_g)| \leq \frac{\Delta_g}{2}$$

implies

$$\Pr\left( \Theta_{t,g} \geq \mu(x_g) + \frac{\Delta_g}{2} \mid \mathcal{F}_{t-1} \right) \leq \exp\left( -2(\eta - 1) \left( \frac{\Delta_g}{2} \right)^2 \right)$$

$$\leq \exp\left( -\frac{c_d n \Delta_g^2}{2} \right)$$

when using Lemma 1 with $\varepsilon = \Delta_g/2$ and $\eta \geq c_d n$. Similarly, for the optimal cell $g^*$, on event $E_{g^*}$ with $N_{g^*} = m$,

$$\Pr\left(\Theta_{t,g^*} \leq \mu(x_{g^*}) - \frac{\Delta_g}{2} \mid \mathcal{F}_{t-1}\right) \leq \exp\left(-\frac{c_d m \Delta_g^2}{2}\right)$$

Now, define the "bad event" that could cause selecting $g$

$$B_t(g) := \{\Theta_{t,g} \geq \Theta_{t,g^*}\}$$

If $g$'s sample is below $\mu(x_g) + \Delta_g/2$ and $g^*$'s sample is above $\mu(x_{g^*}) - \Delta_g/2$, event $B_t(g)$ cannot occur because

$$\Theta_{t,g} < \mu(x_g) + \frac{\Delta_g}{2}$$
$$= \mu(x_{g^*}) - \frac{\Delta_g}{2} \qquad\qquad (\Delta_g = \mu(x_{g^*}) - \mu(x_g))$$
$$< \Theta_{t,g^*}$$

Therefore,

$$B_t(g) \subseteq \left\{\Theta_{t,g} \geq \mu(x_g) + \frac{\Delta_g}{2}\right\} \cup \left\{\Theta_{t,g^*} \leq \mu(x_{g^*}) - \frac{\Delta_g}{2}\right\}$$

Applying the union bound on $B_t(g)$ gives

$$\Pr(B_t(g)|N_g = n, N_{g^*} = m) \leq \Pr(E_g^c|N_g = n) + \Pr(E_{g^*}^c|N_{g^*} = m) + \Pr\left(\Theta_{t,g} \geq \mu(x_g) + \frac{\Delta_g}{2} \mid \mathcal{F}_{t-1}\right)$$
$$+ \Pr\left(\Theta_{t,g^*} \leq \mu(x_{g^*}) - \frac{\Delta_g}{2} \mid \mathcal{F}_{t-1}\right)$$
$$\leq 2\exp\left(-\frac{n\Delta_g^2}{8}\right) + 2\exp\left(-\frac{m\Delta_g^2}{8}\right) + \exp\left(-\frac{c_d n \Delta_g^2}{2}\right) + \exp\left(-\frac{c_d m \Delta_g^2}{2}\right)$$

Since $m \geq 0$, the terms involving $m$ only help. To upper bound, we drop them and keep only the $n$-dependent decay

$$\Pr(B_t(g)|N_g = n, N_{g^*} = m) \leq 2\exp\left(-\frac{n\Delta_g^2}{8}\right) + \exp\left(-\frac{c_d n \Delta_g^2}{2}\right) + 3$$

For it to be a useful bound, we want to set a rule where once $n$ exceeds a logarithmic threshold, the probability of selecting $g$ becomes at most $1/T^2$. Let

$$n_g := \left\lceil \frac{16}{c_d \Delta_g^2} \log T \right\rceil$$

Then for all $n \geq n_g$,

$$\exp\left(-\frac{c_d n \Delta_g^2}{2}\right) \leq \exp\left(-\frac{c_d n_g \Delta_g^2}{2}\right)$$
$$\leq \exp(-8\log T)$$
$$= T^{-8}$$

and similarly $\exp(-n\Delta_g^2/8) \leq T^{-2}$ for a suitable constant. Hence, for $n \geq n_g$,

$$\Pr(B_t(g) \mid N_g = n) \leq C_1 T^{-2} \tag{14}$$

for some constant $C_1$ depending only on $d$.

To bound the expected number of times cell $g$ is selected, we split the sum according to whether $N_g(t-1) < n_g$ or $N_g(t-1) \geq n_g$

$$\mathbb{E}[N_g(T)] = \sum_{t=1}^{T} \Pr(g(s_t) = g)$$

$$\leq n_g + \sum_{t=1}^{T} \Pr(g(s_t) = g, N_g(t-1) \geq n_g)$$

On $\{N_g(t-1) \geq n_g\}$, selecting $g$ implies $B_t(g)$. So,

$$\begin{aligned}
\Pr(g(s_t) = g, N_g(t-1) \geq n_g) &\leq \Pr(B_t(g), N_g(t-1) \geq n_g) && (\{g(s_t) = g\} \subseteq B_t(g)) \\
&= \mathbb{E}[\Pr(B_t(g)|\mathcal{F}_{t-1})\mathbb{1}\{N_g(t-1) \geq n_g\}] && (\mathcal{F}_{t-1} \text{ measurable/non-measurable split}) \\
&\leq \mathbb{E}[C_1 T^{-2} \cdot \mathbb{1}\{N_g(t-1) \geq n_g\}] && (\text{result from Equation 14}) \\
&\leq C_1 T^{-2}
\end{aligned}$$

Therefore, with $C_1/T \leq 1$ for $T \geq 3$,

$$\mathbb{E}[N_g(T)] \leq n_g + TC_1 T^{-2} = n_g + C_1 T^{-1} \leq 2n_g$$

and finally substituting the value of $n_g$ gives

$$\mathbb{E}[N_g(T)] \leq C_2 \frac{\log T}{\Delta_g^2}$$

for each suboptimal cell $g$ where $C_2$ only depends on $d$.

To summarize, the cumulative regret relative to the best cell is

$$\begin{aligned}
\mathbb{E}\left[\sum_{i=1}^{T} \mu(x_{g^*}) - \mu(s_t)\right] &= \sum_{g \in \mathcal{G}} \Delta_g \mathbb{E}[N_g(T)] \\
&\leq \sum_{g:\Delta_g > 0} \Delta_g \cdot C_2 \frac{\log T}{\Delta_g^2} \\
&= C_2 \log T \sum_{g:\Delta_g > 0} \frac{1}{\Delta_g} \tag{15}
\end{aligned}$$

Next, we bound the worst case control of the gap-dependent term via an $\varepsilon$-split. For any threshold $\varepsilon > 0$, split cell into: near optimal $\mathcal{G}_{\leq \varepsilon} = \{g : \Delta_g \leq \varepsilon\}$, and clearly suboptimal $\mathcal{G}_{> \varepsilon} = \{g : \Delta_g > \varepsilon\}$. Trivially, regret from near-optimal cells is at most $T\varepsilon$ because each play can be upper bounded by $\varepsilon$ regret.

$$\sum_{g:\Delta_g \leq \varepsilon} \Delta_g \mathbb{E}[N_g(T)] \leq \varepsilon \sum_g \mathbb{E}[N_g(T)] = \varepsilon T$$

For clearly suboptimal cells, using Equation 15,

$$\sum_{g:\Delta_g>\varepsilon} \Delta_g \mathbb{E}[N_g(T)] \leq C_2 \log T \sum_{g:\Delta_g>\varepsilon} \frac{1}{\Delta_g}$$

$$\leq C_2 \log T \sum_{g:\Delta_g>\varepsilon} \frac{1}{\varepsilon}$$

$$\leq C_2 \log T \frac{|\mathcal{G}|}{\varepsilon}$$

$$\leq C_3 \log T \frac{h^{-d}}{\varepsilon} \qquad \text{using Equation 11}$$

for some constant $C_3$.

Combining results from the $\varepsilon$ splits,

$$\mathbb{E}\left[\sum_{t=1}^{T} \mu(x_{g^*}) - \mu(s_t)\right] \leq \varepsilon T + C_3 \log T \frac{h^{-d}}{\varepsilon}$$

Since within one cell, the Lipschitz variation is $O(Lh)$, the grid optimum $\mu(x_{g^*})$ is only meaningful up to $Lh$, so we set $\varepsilon = Lh$. Upon substitution, the cumulative regret relative to the best cell is

$$\mathbb{E}\left[\sum_{t=1}^{T} \mu(x_{g^*}) - \mu(s_t)\right] \leq LhT + C_3 \log T \frac{h^{-(d+1)}}{L} \tag{16}$$

Continuing from Equation 12, we can begin to form the overall cumulative regret bound.

$$\mathbb{E}[R_T] = \sum_{t=1}^{T} \mu(s^*) - \mu(x_{g^*}) + \sum_{t=1}^{T} \mu(x_{g^*}) - \mu(s_t)$$

$$\leq TL\sqrt{d}h + \sum_{t=1}^{T} \mu(x_{g^*}) - \mu(s_t) \qquad \text{from Equation 12}$$

$$\leq C_4 LhT + C_3 \log T \frac{h^{-(d+1)}}{L} \qquad \text{from Equation 16 and } C_4 := 1 + \sqrt{d}$$

What remains is to optimize the bandwidth with a schedule so as to obtain a useful regret bound. By finding the stationary point,

$$f(h) := C_4 LhT + C_3 \log T \frac{h^{-(d+1)}}{L}$$

$$f'(h) = C_4 LT - C_3 \frac{\log T}{L}(d+1)h^{-(d+2)} = 0$$

we obtain the optimal value of $h$

$$h = \left(\frac{\log T}{L^2 T}\right)^{\frac{1}{d+2}}$$

Plugging $h$ into the regret bound gives

$$\mathbb{E}[R_T] \leq CL^{\frac{d}{d+2}} T^{\frac{d+1}{d+2}} (\log T)^{\frac{1}{d+2}}$$

This is a standard nonparametric Lipschitz-type rate $T^{\frac{d+1}{d+2}}$ with a mild $(\log T)^{\frac{1}{d+2}}$ factor. $\qquad \square$

## A.4 Generation of Correlated Arms

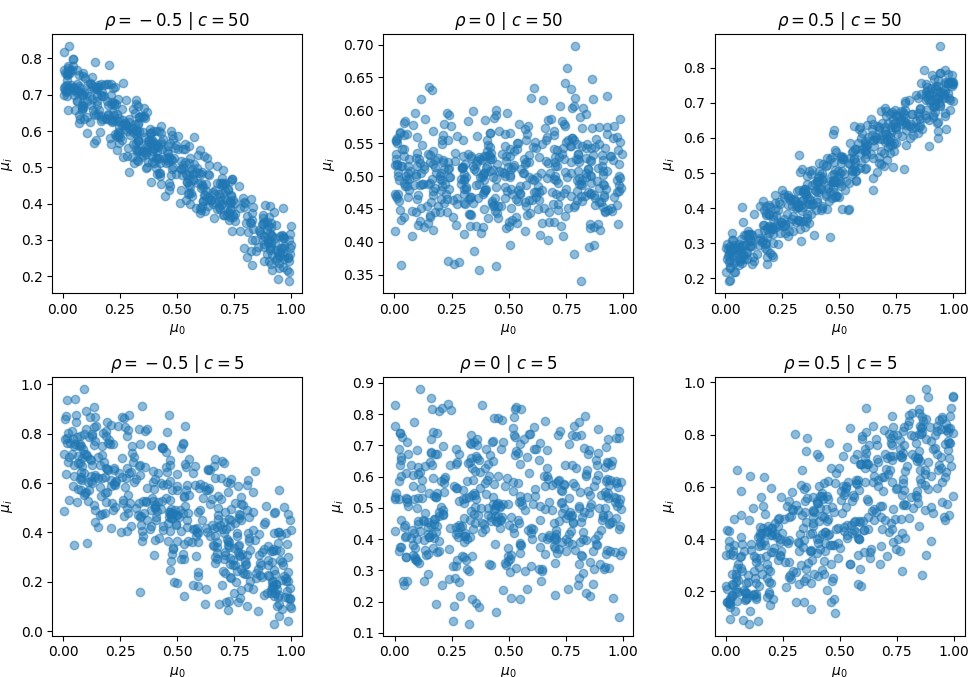

Figure 6: Visualization of impacts of various correlation $\rho$ and $c$ parameters on the $\mu$ of non-anchor arms. Each subplot contains 500 random samples.

Suppose $\mu_0$ has been sampled from the Uniform$(0,1)$ distribution. Now, suppose we want to generate the reward distribution for arm $i$ with a correlation $\rho_i$ with $\mu_0$. The mean parameter for arm $i$ is generated as follows:

$$\mu_i \overset{\text{sample}}{\sim} \text{Beta}(2c(\rho_i(\mu_0 - 0.5) + 0.5), 2c(\rho_i(0.5 - \mu_0) + 0.5)) \tag{17}$$

where $c \geq 1$ is a hyperparameter that controls the variance of the reward distribution of the correlated arm; the higher $c$ is, the lower the variance. In our experiments, the value of $c$ is chosen to be 50. Examples of sampled correlated $\mu_i$'s can be found in Figure 6.

## A.5 Correlation Experiment Details

The training dataset consists of 200 random but correlated values of $\mu_i$'s, and 100 samples of arm-reward pairs with arms being uniformly sampled. $\phi$ is a multilayer perceptron that takes a 7-dimensional vector (one-hot encoded for each arm including the anchor arm), has a single hidden layer of size 256 and an output dimension of 2. Each set of weights is interleaved with a Softplus activation layer. The bandwidth parameter of the RBF kernel is chosen to be $\sigma = 1$.

The loss function for $\phi$ is chosen to be $\mathcal{L}(\hat{\mu}, r) = \mathcal{L}_{\text{BCE}}(\hat{\mu}, r) + 5\mathcal{L}_{\text{ECE}}(\hat{\mu}, r)$, where the ECE loss uses 5 bins. For each of the 4 training epochs, 50% of the entire training dataset is sampled to be used for training, of which 20% will be used as the reference dataset and the remaining 80% contains the queries. The learning rate is $10^{-3}$ (Adam optimizer with default configurations) with an exponential decay rate of 0.99 per epoch. Since no validation dataset is used, the resultant model of the final epoch will be used.

## A.6 Scatter plot of Correlated Arm Embeddings

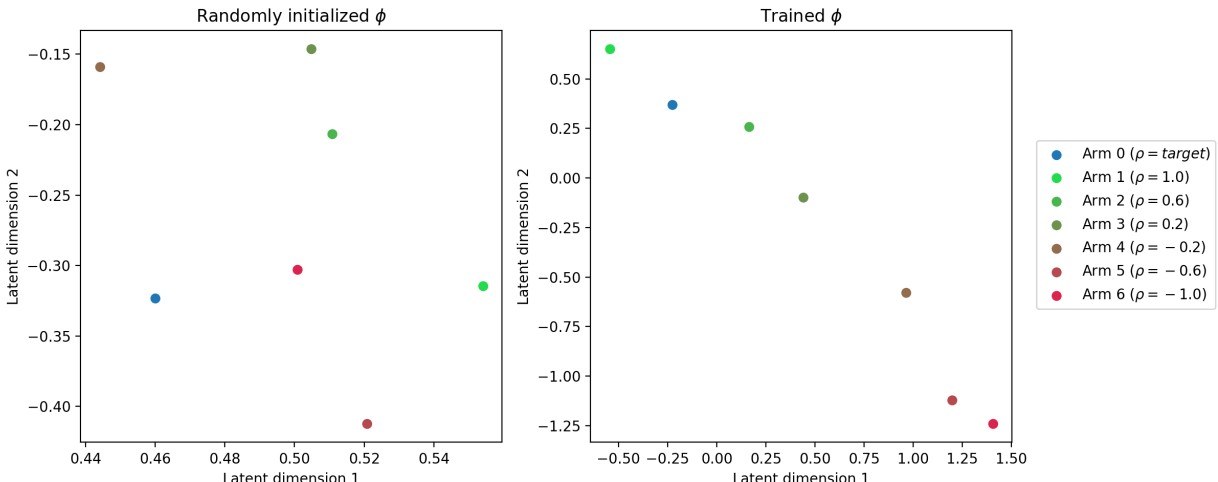

Figure 7: The left figure shows the arm embeddings prior to any fitting while the right figure shows the arm embeddings after training using Algorithm 1. The blue point is the anchor point and the green points are positively correlated to the $\mu$ of the blue point while the red points are negatively correlated.

## A.7 Contextual Bandit Experiment Details

The four datasets were obtained using scikit-learn's API by Buitinck et al. (2013). There is minimal data preprocessing done in this set of experiments: converting the labels to one-hot representations and converting MNIST's pixel values from 8-bit unsigned integers to floating points between 0 and 1.

Only 50% of all datasets were used because of two reasons: (1) using all takes a long time to evaluate especially for NeuralUCB and NeuralTS, (2) the additional 50% during evaluation would only show a longer "linear" portion in the cumulative regret curve since contextual bandit algorithms tend to be unable to practically avoid any mistakes.

All algorithms are given 4 samples to update, followed by 1 evaluation sample (which is also used to update the algorithm). This repeats until 1000 evaluation samples are provided. The datasets are shuffled at the beginning of each of the 10 experiments, so a different 4000 training samples and 1000 evaluation samples are used each time. For the offline training of $\phi$, the training split was used to optimize for $\phi$. This explains why there must be training and evaluation split in this set of experiments.

Let $d_c$ be the dimension of the context vector and $d_a$ be the dimension of the arm vectors. Due to the varying complexities of each dataset, we have to vary the number of layers. Each set of weights is interleaved with a Softplus activation layer. Consider the layers $[\ell_1, ..., \ell_m]$ where $\ell_i$ indicates the size of each layer, and the leftmost and rightmost elements in the array represent the input and output dimensions respectively.

1. shuttle: $[d_c + d_a, 32, 4]$

2. MagicTelescope: $[d_c + d_a, 64, 8]$

3. covertype: $[d_c + d_a, 64, 8]$

4. mnist: $[d_c + d_a, 64, 8]$

The following hyperparameters are the same for all $C_3$ experiments of every dataset. The RBF bandwidth is $\sigma = 1$ and the loss functions $\mathcal{L}_{\text{BCE}}(\hat{\mu}, r) + 2\mathcal{L}_{\text{ECE}}(\hat{\mu}, r)$ where the ECE loss uses 5 bins. The learning rate is set to $10^{-3}$ (Adam optimizer with default configurations) with an exponential decay rate of 0.99 per epoch.

The batch size is 16. During each epoch, 10% of the entire training split is sampled to be used for training, of which 20% will be used as the reference dataset and the remaining 80% contains the queries. There is no partitioning of the reference dataset since this is a stationary problem.

The implementation of LinUCB and LinTS was obtained from a package called `striatum`, while the implementation of NeuralUCB and NeuralTS was obtained directly from the GitHub repository of the authors. SquareCB was obtained from Coba, which uses a linear model as a regression oracle. In LinUCB, we selected $\alpha$ to be 1.96 which controls the exploration factor. In LinTS, we select the default configurations with $\delta = 0.5$ and $R = 0.01$ which are the parameters used in the theoretical regret analysis. For epsilon, it was set to the reciprocal of the number of steps which is the recommended value. For NeuralUCB, we followed the exact hyperparameters that were used in their paper which is $\nu = 10^{-5}$ and $\lambda = 10^{-5}$ (Zhou et al., 2020). For NeuralTS, we set $\nu = 10^{-5}$ and $\lambda = 10^{-5}$ which is obtained from Zhang et al. (2021)'s repository. The update schedule for NeuralUCB and NeuralTS is as follows: for the first 2000 steps, the gradient descent optimization is performed for every step. Afterwards, it is performed only once every 200 steps.

### A.8  Scatterplot of MNIST Context-Arm Embeddings

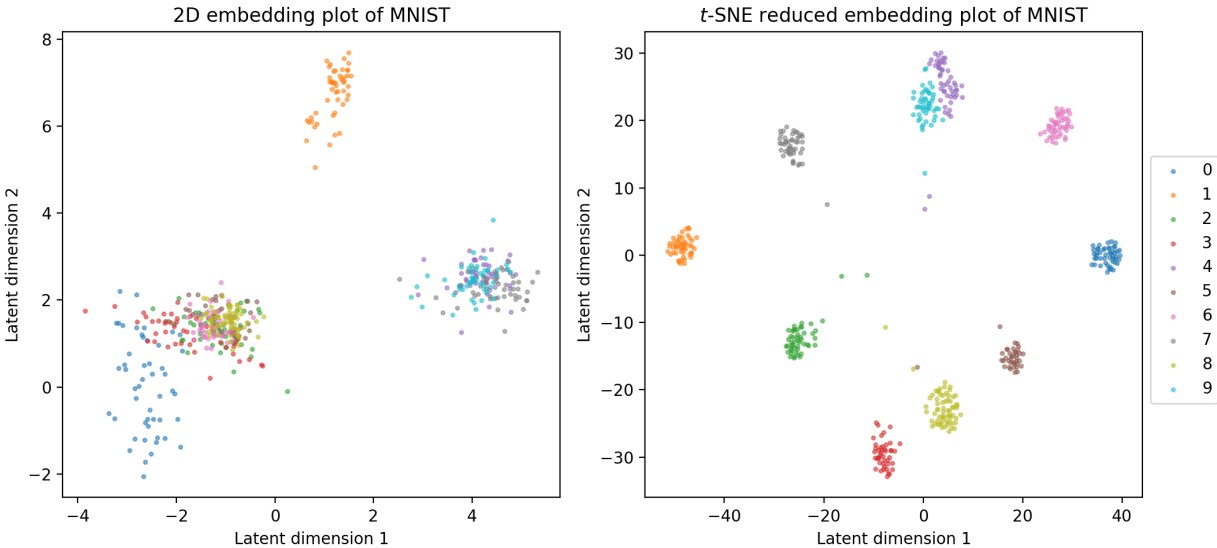

Figure 8: Embedding space of MNIST digits with correct arms chosen where the left shows the embedding vectors of $\phi$ with an output dimension of 2, and the right shows the embedding vectors of another $\phi$ with an output dimension of 8 but uses $t$-SNE (Van der Maaten & Hinton, 2008) for visualization.

## A.9 Ablation on RBF Bandwidth in $C_3$

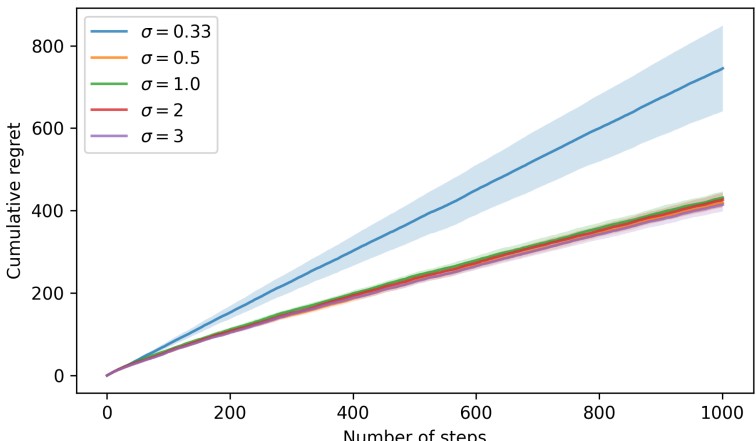

Figure 9: Cumulative regret of $C_3$ with varying $\sigma$ parameter (for the RBF kernel) on the Cover dataset with 1.96 sigma error bars over 10 random seeds.

To demonstrate the effects of varying the RBF kernel, we conduct an ablation study to identify: (1) possible failure cases, and (2) potential robustness across parameters. In experiments in the main paper, the RBF bandwidth $\sigma$ was set to $\sigma = 1$. We vary $\sigma$ by 2 and 3 times larger and smaller, leading to 4 new combinations: $0.33, 0.5, 2, 3$. There are two groups of results in Figure 9. The obvious outlier is $\sigma = 0.33$ which essentially could not learn – a failure case. We postulate this is because the effective "search" radius is too small and did not leverage (or condition on) neighbouring points to make an estimate. The other group, the rest of the $\sigma$'s, showed invariance to the choice of $\sigma$ as they are statistically identical in performance. This is an advantage as this simplifies the hyperparameter tuning process as long as $\sigma$ is not too small.

## A.10 Hardware Usage for Experiments

$C_3$, NeuralUCB and NeuralTS use GPU-acceleration since they have many parameters that need to be optimized. LinUCB, LinTS and SquareCB use CPU only. The GPU used for all experiments in this paper was NVIDIA T4V2 with 16 GB of VRAM. 12 CPU cores with 48 GB of RAM were used for all experiments.

## A.11 MIND Experiment Details

Prior to the experiments, the titles of all news articles are converted to BERT embeddings. The specific pretrained model for BERT from Huggingface (Wolf et al., 2019) is `bert-base-uncased`. The resultant vectors that summarize the entire news article are of size 768 so we used PCA to downsample to 64 dimensions and saved the embedding vectors as a file.

The test dataset provided by MIND does not include labels (Wu et al., 2020) so we could not use that split. Instead, we combine the training and validation datasets that span 7 days. These datasets contain information on article impressions and clicks, i.e. what the users see and which ones do they interact with. Since there are many of them, we decided to be selective and remove instances with nine or more articles shown and sample 20% of the entire dataset. This limits the initial amount of information given to the models and forces the evaluation to be based on incremental learning and exploration strategies.

The first three days were selected to be used as training and validation datasets – 80% training and 20% validation. This implies that reference datasets during the training of $\phi$ will be partitioned into $T = 3$ partitions. The remaining four days will be used as a pool of test data. Given computational constraints, especially over 10 random seeds, we pick only 500 points to be tested. However, the chosen 500 points are

Table 2: Augmenting and diminishing probability schedules by day

| Day | Probability |
|-----|-------------|
| 9   | 0.00        |
| 10  | 0.00        |
| 11  | 0.10        |
| 12  | 0.15        |
| 13  | 0.20        |
| 14  | 0.25        |
| 15  | 0.30        |

selected in a way that takes into account the date. Specifically, every $j^{\text{th}}$ (fixed) entry is selected as test points such that the first and last points are close to the start of the fourth day and end of the seventh day respectively.

To demonstrate stronger concept drift so that its effect can be seen in fewer test samples, we gradually modify the click rate so that the sports category will have a higher click rate over time and all other categories will have a lower click rate in the same time span. An optimal agent should recognize this change and adapt to it.

We augment the click rates of the sports category and diminish the click rate of every other category. For augmenting, for any sports instance which is not a click, we sample a Bernoulli random variable (1 means click, 0 means no click) with probability $\Pr(X = 1) = p_i$ and assign that value to be click value. Similarly, for diminishing, for any non-sports instance which is a click, we sample a Bernoulli random variable with probability $\Pr(X = 0) = p_i$ and assign that value to be the click value. The probabilities by day are shown in Table 2.

$\phi$ was trained with the loss function $\mathcal{L}(\hat{\mu}, r) = \mathcal{L}_{\text{BCE}}(\hat{\mu}, r) + 0.01 \mathcal{L}_{\text{ECE}}(\hat{\mu}, r)$, where the ECE loss uses 5 bins. $\phi$ is initialized to be a multilayer perceptron with input dimension of 82 (18 news categories for context and 64 dimensional arm features), a single hidden layer of dimension 512 and an output dimension of 128. The bandwidth for the RBF kernel is chosen to be 0.6. $\phi$ is trained for 20 epochs with a batch size of 32. The learning rate is set to $10^{-3}$ (Adam optimizer with default configurations) with an exponential decay rate of 0.9. At each epoch, only 10% of the training data is used for training, where 1% of them are used as reference samples while the rest are used as query points. The $\phi$ chosen for testing is the one that attains the lowest validation loss.

For the Bayesian linear regression model, the $\alpha$ parameter, which controls the degree of exploration through the coefficient of the standard deviation of reward, is set to 1.96. $\lambda$, the parameter for numerical stability and regularization, and $\sigma$, the standard deviation of the residuals, are set to 1.

Two towers refer to the context encoder and arm encoder, which are both mappings to the same embedding space. The dot product between the context vector and arm vector represents the logits which are then passed to a softmax layer. The two towers implementation has two variants: small and large. For the small variant, the context encoder layers are $[18, 64, 32]$ and the arm encoder layers are $[64, 128, 32]$. For the large variant, the context encoder layers are $[18, 64, 64]$ and the arm encoder layers are $[64, 256, 64]$. The linear layers of both variants are interleaved with a ReLU activation layer. The small variant was trained for 5 epochs while the large variant was trained for 40 epochs. The learning rates are $8 \times 10^{-3}$ (Adam optimizer with default $\beta_1, \beta_2$) and the batch sizes are 32. For each variant, the model chosen for testing is the one that attains the lowest validation loss.

The modifications to the linear baseline algorithms are sliding window (SW) and discounting (D). Sliding window means that they are refitted with the most recent subset (50000 samples), while discounting uses a weighted linear regression Russac et al. (2019) with a $\gamma$ of 0.95. For the neural baselines, the online modification is changing the buffer into a sliding window (2000 samples). Note that the neural models are initially trained with the entire training size, as the sliding window buffer is only used during test time.

The Gaussian process, due to its cubic time complexity with respect to the number of samples, is restricted to a subsampling of 0.05. The forgetting element works by replacing the oldest sample with the newly seen sample, keeping the buffer the same size. The contextual restless bandit (CRB) requires heavy modification. Since the arms vary with almost no repeating arms, the state space is incredibly sparse and a transition dynamic that is unknown with minimal samples. Our problem does not have a complex budget constraint so the linear programming component is ignored.

## A.12   MIND Time Plot

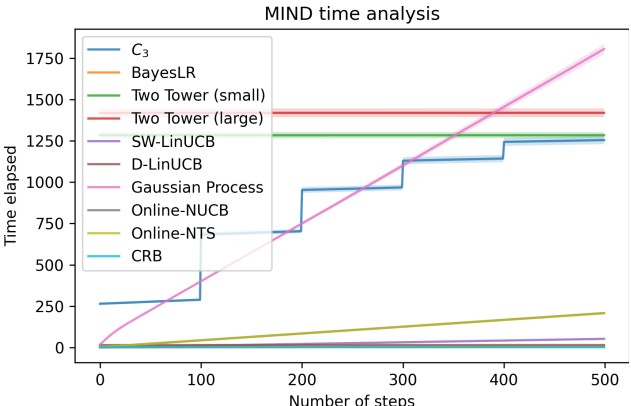

Figure 10: Time taken in seconds against timesteps for each algorithm.

Figure 10 shows a measure of time efficiency for each algorithm. The plot for some algorithms do not start at zero as they require some amount of warmup. The most prominent is the two tower approaches as they are relatively large models. The stepwise pattern in $C_3$ is attributed to an unoptimized approach of dropping samples in its buffer. As mentioned in Section 5.3, 20% of its samples are dropped every 100 steps. This implementation could be improved in future work.

