# OpenReview forum: "A Practical Algorithm for Feature-Rich, Non-Stationary Bandit Problems"
_TMLR — Accepted by TMLR_

### Review · Reviewer_7iGu · 2025-11-08

**Summary Of Contributions:**

This paper proposes a practical Thompson-sampling approach for feature-rich, non-stationary contextual bandits using
(1) learned embeddings, (2) importance-weighted kernel regression, (3) Beta mapping to enable TS without online backprop.

 Strengths
- Practical, deployable design (append/prune ref set; no online gradients).
- Clear method; IW-kernel + Beta posterior is neat.
- Competitive results on OpenML; adapts under synthetic drift on MIND.

**Additional Comments:**

Weaknesses
- Missing drift-aware baselines (sliding-window/discounted LinUCB/TS, online NeuralUCB/TS, kernel/GP with forgetting).
- No regret guarantees (stationary or drifting); theory limited to consistency and $O(n)$ updates.
- “Coupling” notion not learned/quantified; little analysis vs. coupling strength.
- Scalability not validated ($O(kn)$ inference; no ANN/coreset study).

**Audience:**

Yes

**Audience Explanation:**

The paper presents a practical, deployable bandit approach for feature-rich, non-stationary settings (e.g., recommenders) that enables TS without online backprop. Even if evidence is incomplete, these ideas are of clear interest to practitioners and researchers working on real-time decision systems.

**Broader Impact Concerns:**

N.A

**Claims And Evidence:**

No

**Claims Explanation:**

Several central claims are not convincingly supported. Key drift-aware baselines (e.g., sliding-window/discounted LinUCB/TS, online NeuralUCB/TS, kernel/GP with forgetting) are missing; the notion of “coupling” is not measured; there are no regret guarantees for stationary or drifting regimes; scalability ($O(kn)$ inference) isn’t validated; and reproducibility details (code/configs) are incomplete. These gaps make the evidence insufficient to fully support the paper’s claims.

**Requested Changes:**

See additional comments,

---

> ### Author Response · Authors · 2026-01-09
> **Reply to Reviewer 7iGu**
>
> Thank you for the review. Based on your comments, we have included six additional baselines for the MIND experiment (of which five are drift-aware, based on your suggestions). Please see Figure 5. To address some concerns with the lack of regret guarantees, we present a regret bound of $O\left(L^\frac{d}{d+2} T^\frac{d+1}{d+2} (\log T)^\frac{1}{d+2}\right)$ for a stationary $C_3$ case. Please see Appendix A.3 for the proof. As for scalability, we provide an empirical measure of speed in Figure 10.

---

### Review · Reviewer_cc5g · 2025-11-17

**Summary Of Contributions:**

### New Problem Setting
This paper introduces a new contextual bandit framework that combines non-linear reward functions and non-stationary reward distributions, with additional assumption that the arms can be conditionally coupled. This new problem setting is realistic in many practical applications.

### Introduction of C3 Thompson sampling Algorithm
This paper proposes the C3 Thompson sampling algorithm to solve the contextual bandit problem with coupled arms, which is the the first algorithm in this non-stationary setting. By leveraging Importance Weighted Kernel Regression (IWKR) as a nonparametric reward estimator, the algorithm avoids gradient-based updates during inference, improving the efficiency of the method.

### Empirical Evaluation with Practical Datasets
This paper use experiments on real-world datasets to demonstrate the superiority of the performance of the C3 Thompson sampling algorithm. Compared to baseline algorithms such as NeuralTS and NeuralUCB, the new algorithm demonstrates lower cumulative regret in the experiments.

**Audience:**

Yes

**Audience Explanation:**

This paper focuses on non-linear, non-stationary contextual bandit problems and proposed new algorithm framework for these settings, which would be of interest to researchers working on contextual bandits, real-world RL systems, recommendation algorithms, and adaptive learning under drift. Despite lacking theoretical regret bounds, its focus on bridging algorithmic ideas with practical scalability and the demonstrated strong performance make it a useful contribution for applied ML researchers.

**Claims And Evidence:**

Yes

**Claims Explanation:**

The paper claims that the C3 algorithm can effectively handle dense arm features, non-linear reward functions, and non-stationarity and outperforms the prior state-of-the-art algorithms in regret minimization. These claims are supported by experiments across different real-world datasets and are convincing.

Although the empirical performance of the C3 algorithm is convincing, the authors didn't provide any formal theoretical guarantees of the regret bound and how it compares to baseline algorithms theoretically. Also, the paper could add comparisons to recent non-stationary bandit algorithms such as Restless UCB to strengthen the claim.

**Requested Changes:**

### The following proposed adjustments would strengthen the work but are not critical:
(1) Add recent bandit algorithms specifically designed for non-stationary settings such as Restless-UCB into the baseline comparisons.
(2) In Section 3.2, add extra explanations of coupled arms and how coupling persists over time in non-stationary environments.
(3) Add comparisons of runtime and memory usage between different algorithms to further demonstrate the efficiency of the new algorithm

---

> ### Author Response · Authors · 2026-01-09
> **Reply to Reviewer cc5g**
>
> Thank you for the review. We address your concern on the lack of theoretical guarantees by showing that $C_3$ incurs a regret bound of $O\left(L^\frac{d}{d+2} T^\frac{d+1}{d+2} (\log T)^\frac{1}{d+2}\right)$ for a stationary case. Please see Appendix A.3 for the proof. In addition, we included six new additional baselines for the MIND experiment, one of them being the contextual restless-UCB algorithm. Please see Figure 5. We added more explanation and an example in Section 3.2 on coupling in a non-stationary environment. Finally, we included Figure 10 in Appendix A.12 to show the efficiency of $C_3$ compared to other neural approaches.

---

### Review · Reviewer_23th · 2025-12-08

**Summary Of Contributions:**

This work studies the learning problem of contextual bandits with correlated arms in a non-stationary reward setting. They devised a conditionally coupled contextual Thompson sampling (C3) algorithm, evaluated its empirical performance with some classical algorithms such as LinUCB and LinTS, and demonstrated that its empirical performance outperforms the compared algorithms.

**Additional Comments:**

I also have some questions.

1. For problem set-up in Section 3.1, the key feature is the Bernoulli mean parameter function $\mu: \mathcal{C} \times \mathcal{A} \rightarrow [0,1]$ is assumed to be Lipschitz continuous. If I assume that the context for each arm $a \in \mathcal{A}$ is a Lipschitz continuous function, can you clarify the difference between your setting and the setting I have mentioned?

2. What is the key technical difficulty between the stationary and non-stationary reward setting, i.e.,
$\mu: \mathcal{C} \times \mathcal{A} \rightarrow [0,1]$ vs $\mu: \mathcal{C} \times \mathcal{A} \times [T] \rightarrow [0,1]$ in terms of developing algorithms?

**Audience:**

Yes

**Audience Explanation:**

The same as above.

After reading the rebuttals:

I do believe and am convinced that practitioners will be more interested in this work.

**Claims And Evidence:**

No

**Claims Explanation:**

1. The literature discussion is not sufficient. All the references in this work are pretty out-of-date. The most recent cited work is from 2023. The majority of the references are around 2020. For the past 5 years, there have been plenty of learning algorithms for solving contextual bandit and Gaussian process bandit problems.

2. This work does not provide any meaningful theoretical results.

3. Empirically, C3 algorithm is not compared to the state-of-the-art algorithms. Also, as compared to LinTS, I suspect the empirical performance improvement is from the choice of priors. Usually, we have beta prior gives better performance than Gaussian priors. So, it is not a fair comparison.

**Requested Changes:**

1. I suggest to adding more recent literatures.

2. Empirical experiments also need to use the state-of-the-art algorithms as the baseline.

---

> ### Author Response · Authors · 2025-12-16
> **clarification regarding state of the art algorithms**
>
> Thank you for the review and the feedback.  You made a strong point about the need to compare to state of the art algorithms.  Can you clarify what state of the art algorithms are missing in our baselines?

---

> ### Author Response · Authors · 2026-01-09
> **Reply to Reviewer 23th**
>
> Thank you for the review. Based on your comments, we added two more recent related works, both from 2024. To address the lack of theoretical results, our rebuttal version of the paper now includes a proof of regret bound of $O\left(L^\frac{d}{d+2} T^\frac{d+1}{d+2} (\log T)^\frac{1}{d+2}\right)$ for a stationary case. Please see Appendix A.3 for the proof. Regarding the empirical performance of LinTS versus $C_3$, it is a deliberate decision to focus on Beta priors because the scope of our work is on Bernoulli bandits as mentioned in Section 3.1. Finally, we have yet to receive a reply to our message on December 16, but to address your concerns about state-of-the-art baselines, we added six new baseline algorithms in Figure 5.
>
> We will address your questions here.
> 1. We apologize in the event we misinterpret your first question. The context is given, and is not a function in any way, hence it cannot be a Lipschitz continuous function as mentioned at the start of your second sentence. Also, there is only one context for each time step, not a context for each arm. However, if you perhaps meant that $\mu$ is Lipschitz continuous with respect to the arm $a$ **only** (and not context $c$), then the answer to your question is: theoretically, there is not a significant difference as long as the embedding function/model $\phi$ can learn a sufficiently calibrated metric space. Practically, however, this might lead to difficulty in optimizing $\phi$.
>
> 2. The difficulty of extending from a stationary to a non-stationary algorithm lies in adapting to an underlying shift of the reward function, while having to explore only one arm at a time. One clear direction is to directly re-optimize the parameters of the model with each new sample but since we are concerned with practical, real-time use-cases, this is not feasible due to frequent retraining costs. There is also a problem of tuning the threshold of which samples to discard or retain, since we have no information on how the reward function shifts. Excessively discarding samples increases the probability of discarding useful information, while retaining too many old samples might result in failing to adapt to the changing reward function.

---

### Author Response · Authors · 2025-12-16
**request for extension due to holidays**

Dear Pan Xu.  The first author for our paper returned home for the Christmas holidays, shortly after we received the 3rd review. Unfortunately, this did not leave us enough time to complete some of the additional experiments requested by the reviewers.   While the instructions indicate that we need to revise the paper and submit our rebuttal by Dec 22, would it be possible for us to get an extension till January 10?  This will give the first author a week to complete the revisions including the requested experiments after his return to the university.  Thanks in advance.

---

> ### Comment · Action_Editor_5P9W · 2025-12-17
>
> Dear authors,
>
> We’re pleased to inform you that your request for an extension on the rebuttal deadline has been granted. We expect your responses by January 10 before proceeding to the next stage.
>
> Thanks,
>
> Pan

---

### Decision · Action_Editor_5P9W · 2026-02-06

**Recommendation:** Accept with minor revision

**Additional Comments:**

The algorithm primarily relies on heuristics, and its performance heavily depends on the quality of the embedding space. While the paper focuses on nonstationary bandits, the theoretical results provided are applicable only to stationary bandits, which are well-established in the literature. Stronger theoretical results would enhance the algorithm’s quality and bolster its empirical support. Therefore, the current paper should address and emphasize this limitation in their revision.

**Audience:**

Yes

**Audience Explanation:**

The paper deals with high-dimensional contextual and nonstationary bandits, which is of interest to many audience in this field.

**Claims And Evidence:**

Yes

**Claims Explanation:**

The paper presents a practical algorithm for high-dimensional contextual and nonstationary bandits. Following the rebuttals, additional baselines were incorporated, and the claims are largely supported by experiments conducted on recommendation tasks.

---

> ### Author Response · Authors · 2026-03-06
> **Completion of revision for camera-ready paper**
>
> Dear Pan Xu,
>
> We have incorporated your suggested points in the revision. We summarize the changes below:
> * Expanded the "Limitations of C3" section with mentions of heuristics and dependency on well-calibrated embedding space
> * Renamed the conclusion section to "Conclusion and Future Works" and added future work to extend the bound to a non-stationary case
> * Added acknowledgements, de-anonymized authors, and added OpenReview and GitHub link
> * Rectified minor grammatical error and notations for clarity
>
> Thank you.